# Echinochrome A Attenuates Cerebral Ischemic Injury through Regulation of Cell Survival after Middle Cerebral Artery Occlusion in Rat

**DOI:** 10.3390/md17090501

**Published:** 2019-08-28

**Authors:** Ran Kim, Daeun Hur, Hyoung Kyu Kim, Jin Han, Natalia P. Mishchenko, Sergey A. Fedoreyev, Valentin A. Stonik, Woochul Chang

**Affiliations:** 1Department of Biology Education, College of Education, Pusan National University, Busan 46241, Korea; 2National Research Laboratory for Mitochondrial Signaling, Department of Physiology, College of Medicine, Cardiovascular and Metabolic Disease Center (CMDC), Inje University, Busan 614-735, Korea; 3G.B. Elyakov Pacific Institute of Bioorganic Chemistry, Far-Eastern Branch of the Russian Academy of Science, Vladivostok 690022, Russia

**Keywords:** echinochrome A, brain ischemic stroke, cell survival

## Abstract

Of late, researchers have taken interest in alternative medicines for the treatment of brain ischemic stroke, where full recovery is rarely seen despite advanced medical technologies. Due to its antioxidant activity, Echinochrome A (Ech A), a natural compound found in sea urchins, has acquired attention as an alternative clinical trial source for the treatment of ischemic stroke. The current study demonstrates considerable potential of Ech A as a medication for cerebral ischemic injury. To confirm the effects of Ech A on the recovery of the injured region and behavioral decline, Ech A was administered through the external carotid artery in a rat middle cerebral artery occlusion model after reperfusion. The expression level of cell viability-related factors was also examined to confirm the mechanism of brain physiological restoration. Based on the results obtained, we propose that Ech A ameliorates the physiological deterioration by its antioxidant effect which plays a protective role against cell death, subsequent to post cerebral ischemic stroke.

## 1. Introduction

Although medical knowledge and technology have advanced considerably, cerebral ischemic stroke patients experience pain with severe disabilities and high mortality due to brain damage resulting from middle cerebral artery occlusion (MCAo) [1,2]. The progression of pathological changes in acute ischemic stroke includes oxidative stress, free radical production, brain edema, neuronal apoptosis, and finally death [3,4]. Intravenous recombinant tissue plasminogen activator (rtPA) or other agents such as anti-cytokines, calcium channel blockers, and free radical scavengers for therapeutic thrombolysis are the generally applied treatments in the early stage of the disease [5]. However, there is rarely a complete recovery of ischemia neuronal damage. Therefore, in recent years, researchers have recognized the importance of finding inventive strategies for the restoration of brain injury. 

Natural derived products, including molecules from marine organisms, have gained recognition as alternative therapeutic sources. The past decade has seen a rise in the application of various compounds from isolated marine species as a popular and promising therapeutic approach for human diseases [6]. Marine organisms are capable of overcoming some insuperable conditions such as light, temperature, pressure, and oxygen and ion concentration due to their distinct habitat environment [7]. Echinochrome A (Ech A), a dark red pigment, is most commonly extracted from the shells, spines, and eggs of sea urchins [8]. The biologically active compound possesses antiviral, antialgal, and antioxidant properties [8,9]. Ech A is an active substance in the cardioprotective drug Histochrome, registered in Russia (P N002363/01). It presented a positive outcome in animal disease model with experimental hemorrhagic stroke as previously described [10]. Histochrome accelerates the alleviation of neurological symptoms and edema, which are mainly caused by the release of free iron and oxidative stress. The effects of Ech A on ischemic stroke have not been previously studied. In particular, the high antioxidant efficacy exerts a beneficial advantage in the research of stroke [11]. Oxidative stress contributes to irreversible pathophysiological cellular damages during the initial and later phases of ischemic stroke [11]. Despite the restoration of blood flow to rescue the ischemic brain, reperfusion is known to aggravate oxidative stress damage and generate reactive oxygen species (ROS) [12]. Ech A neutralizes the iron cations that accumulate in the region of ischemic damaged tissue [12]. However, researchers have rarely considered Ech A as a therapeutic candidate to overcome brain ischemic stroke.

Based on the results obtained using an experimental rat MCAo model, we demonstrate the potential of Ech A for treating brain ischemic stroke. In the present study, the therapeutic effects of Ech A were evaluated. The administration of Ech A recovered the brain region and alleviated the repressed behaviors in the rat MCAo model. Our results indicate that the administration of Ech A influences the expression of cell viability related factors, thereby confirming its effect on physiological improvements. 

## 2. Results and Discussion

### 2.1. Ech A Mitigates Cerebral Ischemic Injury

To confirm the mitigative effect of Ech A on cerebral ischemic disease, rat MCAo models were exposed to 10 μM Ech A and subsequently assessed for brain infarct volume and water content. We used 2,3,5-triphenyltetrazolium chloride (TTC) staining, one of the most conventional methods, to visualize the infarct region of the ischemic cerebrum. Ech A substantially shows the visual recovery reducing the brain infarct volumes at 7 days after reperfusion (Figure 1A). Alleviation of the injured area after exposure to Ech A was confirmed by comparing the percentage of white region in the control brain with the Ech A treatment group through TTC staining (Sham: 0.0%, control group: 52.7%, 10 μM Ech A: 21.9%) (Figure 1B). Next, we investigated the water content of rat MCAo model brain tissue, which is a marker of severe ischemic injury [13]. The water content is used to signify the ischemic brain edema in the infarct hemisphere [14], and to examine the effect of Ech A treatment on the blood–brain barrier (BBB) leakage in this study. Cerebral edema plays a key role in fatal outcomes associated with numerous neurological conditions, including ischemic stroke [15]. The percentage of water content in the control group was increased, as compared to the sham group. The Ech A-treated group revealed decreased water content as compared to the control group (sham: 79.4%, control: 81.7%, 10 μM Ech A: 80.3%) (Figure 1C). Taken together, we construe that Ech A presents the visual restoration in the infarct cerebrum of rat MCAo models. However, the functional recovery and the reparative mechanism by Ech A treatment on brain ischemic disease remain to be identified.

### 2.2. Ech A Encourages Affirmative Behavioral Changes after Ischemic Stroke

Functional assessments were applied to determine behavioral changes in the rat model for aggressive central nervous damage [16]. Two methods, namely, the cylinder test and swim test, were adopted and modified. The cylinder test evaluates the frequency of forelimb use and asymmetric movement in postural weight support [17]. In the cylinder test, cerebrally injured rats (control group) exhibited increased asymmetry use, especially practical use of the unaffected forelimb, when compared to the sham group. Furthermore, the 10 μM Ech A-treated group showed behavioral recovery with increased use of both forelimbs and decreased use of the unaffected forelimb, as compared to the control group (simultaneous use: sham 38.0%, control 26.0%, 10 μM Ech A 35.0%; unaffected use: sham 26.0%, control 39.0%, 10 μM Ech A 32.0%) (Figure 2A). 

Stroke results in the reduction of predominant activities such as mobility, climbing, and swimming [18]. Since the forced swim test (FST) is frequently used to confirm these activities, this method was applied to assess the antidepressant-like behavior in MCAo models [18]. In this test, while brain infarct rats showed increasing immobility, the behavioral recovered rats (after 10 μM Ech A treatment) show alleviated immobility time during the FST (sham: 33 s, control: 81 s, 10 μM Ech A: 51 s) (Figure 2B). Our results indicate that treatment with an appropriate amount of Ech A enhances the motor ability in brain ischemic disease.

### 2.3. Ech A Affects the Expression of Cell Survival-Related Molecules of Rat Ischemic Stroke Brain

To demonstrate the mechanism of physiological improvements, which include the restoration of the damaged brain region and the intensification of attenuated behavior after Ech A treatment, we focused on the occurrences after ischemia reperfusion injury. Ischemia reperfusion-injured brain suffers from oxidative stress and induces the cell death regulating pathway [19]. Considering this, we investigated the expression levels of cell viability-related factors in our experimental animal model, including Bcl-2, Caspase-3, Bax (Figure 3A), p-ERK/ERK, p-AKT/AKT (Figure 3B), and brain-derived neurotrophic factor (BDNF) (Figure 3C). The effect of Ech A treatment on the expression of these regulators was estimated in the brain tissue of MCAo rats. Bcl-2, caspase-3, and Bax work as major mediators for cell survival and death and are activated by various stimuli [18,19]. Bcl-2, an apoptosis inhibitor, is a key player in the mechanism of anti-apoptosis [20,21]. In contrast, caspase-3 and Bax are pro-apoptosis molecules which signify the onset of apoptosis [21]. Compared to the control group, Ech A treatment in the MCAo rat model significantly increased the expression level of Bcl-2 and decreases the levels of caspase-3 and Bax. The extracellular signal-related kinases (ERK) are essential regulators associated with vital cellular functions, including cell proliferation, differentiation, migration, senescence, and apoptosis in the generic mitogen-activated protein kinase (MAPK) signaling pathway [22]. Furthermore, in the PI3K/AKT/mTOR signaling pathway, AKT is also a core component of various processes of cellular activities, including nutrient uptake, anabolic reactions, metabolism, cell growth, proliferation, differentiation, apoptosis, and survival [23]. Our results indicate an increase in the expression levels of p-ERK/ERK and p-AKT/AKT in the Ech A-treated MCAo rat model as compared to the control group. The brain-derived neurotrophic factor (BDNF) significantly supports neuronal differentiation and survival, synaptic formation and plasticity, and neurogenesis, and has been widely researched in various neurological conditions [24]. Our studies reveal increased BDNF expression in the brain of MCAo rat model after exposure to Ech A, as compared to the control group. Taken together, these findings confirm that Ech A relieves the physiological decline in the MCAo rat model by increasing and supporting cell survival in the injured brain region.

## 3. Experimental Section

### 3.1. Chemical

Ech A (6-ethyl-2,3,5,7,8-pentahydroxynaphthalene-1,4-dion) was isolated from the sand dollar *Scaphechinus mirabilis* using an extraction method as previously described [25]. The purity of Ech A (>99%) was confirmed by liquid chromatography-mass spectrometry (Shimadzu LCMS-2020, Kyoto, Japan). We used 0.02% Ech A with saline solution.

### 3.2. Preparation of Ischemic Stroke Rat Models 

#### 3.2.1. Animals

Nine-week-old male Sprague Dawley (SD) rats (290–300 g; KOATECH, Gyeonggi-do, Korea) were handled in accordance with the animal welfare guidelines issued by the Korean National Institute of Health and the Korean Academy of Medical Sciences for the care and use of laboratory animals.

#### 3.2.2. Middle Cerebral Artery Occlusion

We applied a modified surgical procedure of the standard method [26,27]. Briefly, rats were anesthetized with isoflurane in a mixture of 30% oxygen and 70% nitrous oxide. Using an operative microscope, a 3-0 nylon suture was inserted into the internal carotid artery (ICA) through the external carotid artery (ECA). After 90 min of occlusion, the suture was withdrawn for 7 days for reperfusion. The animals were randomly divided into three groups (*n* = 12 per group): group I, sham-operation (sham), group II, MCAo/reperfusion-induced ischemic group with saline treatment (control), group III, MCAo/reperfusion-induced ischemic group with 10 μM Ech A treatment.

### 3.3. Behavioral Test

We modified the FST and cylinder test to determine the motor ability of experimental animals. For the FST, at 7 days after reperfusion, rats were placed in an open cylinder (height: 60 cm, diameter: 20 cm) which was filled with water for 6 min, and their duration of immobility was measured for the last 4 min. In the cylinder test, rats were placed in a transparent cylinder (height: 30 cm, diameter: 20 cm) for 6 min. The use of their forelimbs was recorded for the last 4 min. The number of right and left forelimbs used independently, and both forelimbs use simultaneously, were observed and recorded. Each experiment was repeated five times.

### 3.4. Measurement of Brain Infarct Volume and Water Content

All animals were euthanized 7 days after MCAo operation. Brains were collected and sectioned into 2.0 mm coronal slices to assess the brain infarct volume. The brain sections were stained with TTC and the percentage of infarct volume was calculated by assessing the stained brain area using ImageJ software. To measure the brain water content, the pons and olfactory bulbs were removed and the wet weight (ww) of the brain was measured. The brains were subsequently dried at 110 °C for 24 h, and the dry weight (dw) of the brain was examined. The percentage water content in the brain was assessed using the following formula:Water content: (ww − dw)/ww × 100%

### 3.5. Western Blot

Brain tissues were homogenized in lysis buffer (Cell Signaling Technology, Beverly, MA, USA) to collect the total proteins. Quantified proteins were separated by SDS-PAGE and transferred to polyvinylidene fluoride microporous membrane (Millipore, Temecula, CA, USA). The membranes were blocked with 0.1% Tween 20 in Tris-buffered saline containing 5% nonfat milk for 1 h at room temperature, and subsequently incubated with the primary antibody (Bcl-2, caspase-3, Bax, p-ERK, and ERK; Santa Cruz Biotechnology, Dallas, TX, USA; p-Akt, Akt, and β-actin; Cell Signaling Technology). 

### 3.6. Polymerase Chain Reaction (PCR) Analysis

The total RNA of brain tissues was extracted using the Hybrid-R RNA purification kit (GeneAll, Seoul, Korea), and converted to cDNA using the cDNA synthesis kit (Thermo Scientific, Vilnius, Lithuania). Amplification was performed in a DNA thermal cycler using the following synthesized primers: 5′-GATGAGGACCAGAAGGTTCG-3′ (forward) and 5′-GATTGGGTAGTTCGGCATTG-3′ (reverse) for BDNF; 5′-GCTGGGGCTCACCTGAAGGG-3′ (forward) and 5′-GGATGACCTTGCCCACAGCC-3′ (reverse) for GAPDH.

### 3.7. Statistical Analysis

All experimental data are presented as mean ± standard error of the mean (SEM). Comparisons between more than two groups were performed by one-way ANOVA using Bonferroni’s correction. A *p*-value < 0.05 is considered significant.

## 4. Conclusions

Until recently, Ech A was rarely used for the treatment of brain ischemic stroke. Our findings indicate that Ech A as a novel therapeutic source from the ocean has considerable efficacy for cerebral ischemic injury. We demonstrate that Ech A restores the damaged brain area and strengthens the behavioral deterioration by supporting the expression of cell viability-related factors after brain ischemic stroke. Taken together, the results of this study propose a new application as a potential therapeutic agent for this marine drug.

## Figures and Tables

**Figure 1 marinedrugs-17-00501-f001:**
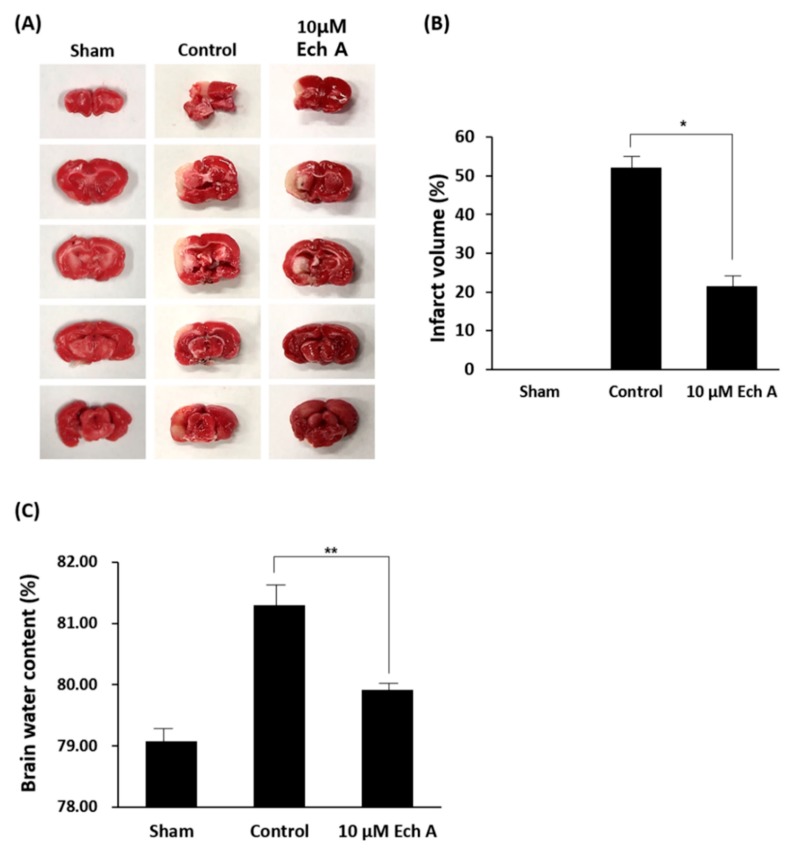
Echinochrome A (Ech A) alleviates the infarcted brain region of rat middle cerebral artery occlusion (MCAo). (**A**) Representative TTC staining of brain sections of the damaged area in the sham, control and 10 μM Ech A treatment groups of a rat MCAo/reperfusion model. (* *p* < 0.05). (**B**) Quantification of size in the infarcted brain region from each experimental group. (**C**) Quantification of water content in infarcted brain region from each experimental group (** *p* < 0.01).

**Figure 2 marinedrugs-17-00501-f002:**
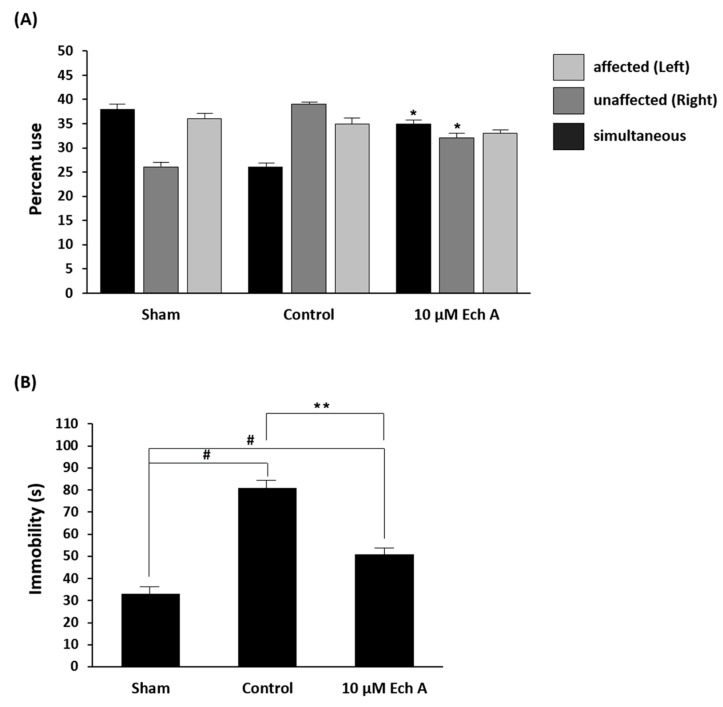
Declined movements are restored after Ech A treatment following ischemic stroke. (**A**) Assessment of percent use of affected (left), unaffected (right), and simultaneous (both) forelimbs on the wall of the cylinder (* *p* < 0.05). (**B**) Total amount of immobility time in the forced swim test (^#^
*p* < 0.01 compared with the values of the sham group; ** *p* < 0.01 compared with the values of the control group).

**Figure 3 marinedrugs-17-00501-f003:**
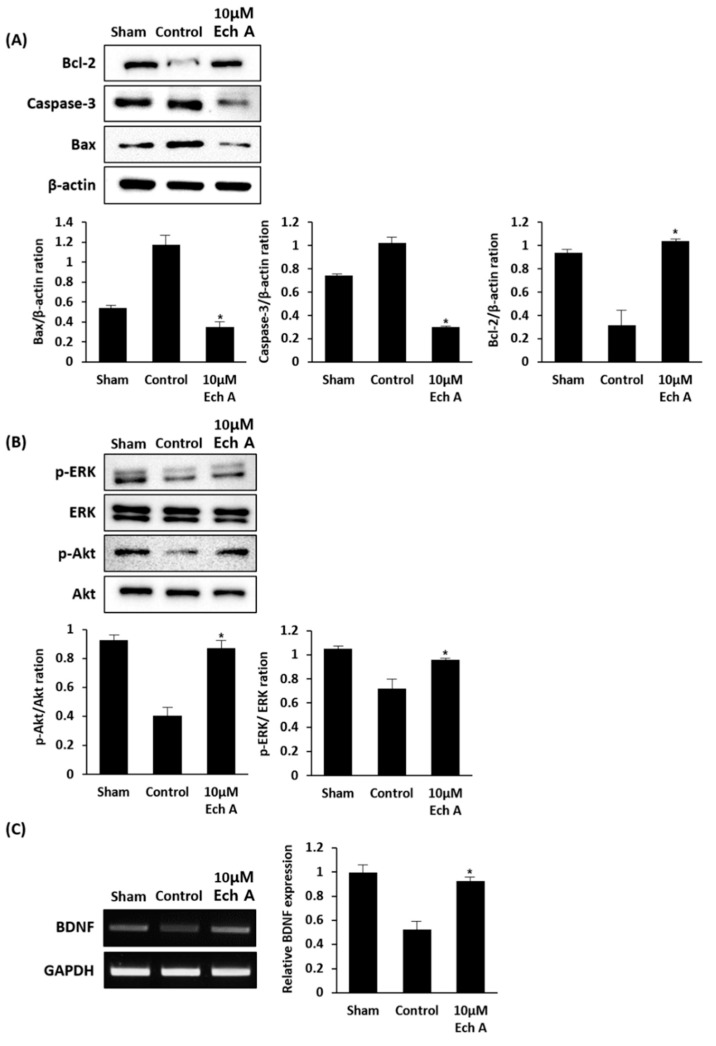
Ech A treatment in cerebral injured brain alters the expression levels of cell viability-related factors. (**A**) The protein expression levels of cell survival and death regulators, such as Bcl-2, caspase-3, and Bax. (**B**) The protein expression levels of key players in vital cellular function regulation pathways, such as ERK and AKT. (**C**) The mRNA expression level of BDNF, supporting cell survival alteration in the injured brain region (* *p* < 0.05).

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
