# Peer review of "Echinochrome A Attenuates Cerebral Ischemic Injury through Regulation of Cell Survival after Middle Cerebral Artery Occlusion in Rat"

_marinedrugs, 2019, doi:10.3390/md17090501_

Round 1

Reviewer 1 Report

This one is an interessting paper.

My reccomendation is to separe results and discussion

Author Response

Point 1: My reccomendation is to separe results and discussion.

Response 1: Thank you for your careful recommendation. We wrote this paper based on the previous published paper in Marine Durgs [1]. Furthermore, we are so sorry that there is not enough time to separate in two parts, Results and Discussion.

[1] Jeong, S.H.; Kim, H.K.; Song, I.S.; Noh, S.J.; Marquez, J.; Ko, K.S.; Rhee, B.D.; Kim, N.; Mishchenko, N.P.; Fedoreyev, S.A.; Stonik, V.A.; Han, J. Echinochrome a increases mitochondrial mass and function by modulating mitochondrial biogenesis regulatory genes. Mar. Drugs. 2014, 12, 4062-4615.

Reviewer 2 Report

Section 2 Results and discussion

The authors must better explain section 2.1

Page 2 line 70 EchA ....... 

The authors have reported a decrease in the water content with respect to the control…. The reduction is not significant. Authors need to better explain the sentence

Section 3. Experimental section 3.2.2 Occlusion of the middle cerebral artery.

The authors must explain why they chose 90 min of ischemia and 7 days of reperfusion

The authors must report the required reperfusion times and whether it was done in a time course of ischemia and reperfusion times

The authors must clarify whether the same reperfusion time (7 days) was used for all experiments

Authors must correct the English language

Author Response

Section 2 Results and discussion

Point 1: The authors must better explain section 2.1

Response 1: Thank you for your advice. The suggestion has been well taken. We added more description for section 2.1 (page 2 line 65-81).

Point 2: Page 2 line 70 EchA .......

The authors have reported a decrease in the water content with respect to the control…. The reduction is not significant. Authors need to better explain the sentence

Response 2: Thank you for your comment. We constructed our experiments based on other researchers' papers. Gandin et al. (2016) demonsted that water content was used to invastigate ischemic brain recovery with in farct volume measurements, the experimental group showing significantly reduced brain edema when compared to vehicle group. And their treated group significantly decreased the brain water
content compared to the vehicle group [1]. And Chen et al. (2015) also showed that brain water content was reduced in their treated
group when compared with the PBS-treated group [2]. However, we added more description for the part of water content according to your advice (page 2 line 73-75).

[1] Gandin. C.; Widmann, C.; Lazdunski, M.; Heurteaux, C. MLC901 Favors Angiogenesis and Associated Recovery after Ischemic Stroke in Mice.  Cerebrovasc. Dis. 2016, 42, 139-154.
[2] Chen, M.; Li, X.; Zhang, X.; He, X.; Lai, L.; Liu, Y.; Zhu, G.; Li, W.; Li, H.; Fang, Q.; Wang, Z.; Duan, C. The inhibitory effect of mesenchymal stem cell on blood-brain barrier disruption following intracerebral hemorrhage in rats: contribution of TSG-6. J. Neuroinflammation. 2015, 12, 61. doi: 10.1186/s12974-015-0284-x.

Section 3. Experimental section 3.2.2 Occlusion of the middle cerebral artery.

Point 3:  The authors must explain why they chose 90 min of ischemia and 7 days of reperfusion

Response 3: Thank you for your advice. The experimental method which we used in the present paper was already stabilized in our laboratory based on the papers by Longa et al. (1989) and Hill and Nemoto (2014) [1,2]. Furthermore, our previous studies were pulished using this method [3,4]. So We stated a sentence for section 3.2.2, 'We applied a modified surgical procedure of the standard method [24, 27]'.

[1] Longa, E.Z.; Weinstein, P.R.; Carlson, S.; Cummins, R. Reversible middle cerebral artery occlusion without craniectomy in rats. Stroke 1989, 20, 84-91.
[2] Hill, J.W.; Nemoto, E.M. Transient middle cerebral artery occlusion with complete reperfusion in spontaneously hypertensive rats. MethodsX 2014, 1, 283-291.
[3] Kim, R.; Lee, S.; Lee, C.Y.; Yun, H.; Lee, H.; Lee, M.Y.; Kim, J.; Jeong, J.Y.; Baek, K.; Chang, W.Salvia miltiorrhiza enhances the survival of mesenchymal stem cells under ischemic conditions. J. Pharm. Pharmacol. 2018, 70, 1228-1241.
[4] Kim, R.; Kim, P.; Lee, C.Y.; Lee, S.; Yun, H.; Lee, M.Y.; Kim, J.; Baek, K.; Chang, W. Multiple Combination of Angelica gigas Extract and Mesenchymal Stem Cells Enhances Therapeutic Effect. Biol. Pharm. Bull. 2018, 41, 1748-1756.

Point 4: The authors must report the required reperfusion times and whether it was done in a time course of ischemia and reperfusion times

Response 4: Thank you for your advice. The suggestion has been well taken (page 4 line 141).

Point 5: The authors must clarify whether the same reperfusion time (7 days) was used for all experiments

Response 5: Thank you for your advice. The suggestion has been well taken.

Point 6: Authors must correct the English language

Response 6: Thank you for your comment. We attached the certificate of proofreading and editing by English editor.

Reviewer 3 Report

In this work, Prof. Woochul Chang and collaborators show that direct injection of EchA through ECA (external carotid artery) injection to MCAo rat model reduced infarct volume. Furthermore, EchA injection improved functional behavioral recovery of the animals with significant increase in cell viability factors. These finding indicate that EchA has potential as a treatment for hypoxic-ischemic brain injury and as it is an interesting study it would be recommended for additional experiments to further support their hypothesis.

Major comments

(i)            Since the author explain the main mechanism is driven by reducing the cell death regulation pathway it would be important to quantify the ischemic-induced apoptosis from the infarcted brain region and show the effect of EchA form the same region. I recommend measuring the fragmentation of nuclear DNA of the penumbral area by TUNEL (terminal deoxynucleotidyl transferase dUTP nick-end labeling).

Minor comments

(i)            Typing error in 3.2.2 Middle Cerebral Artery Occlusion

External carotid artery (CCA) à External carotid artery (ECA)

(ii)           In figure 2. A, it is hard to visualize a difference between affected, unaffected and simultaneous in different groups. I recommend the percentage typed in each column so each column can be better compared.

(iii)          Figure 2, legend need indication of # and **.

(iv)          In Figure 3, each western and northern blot should have an indication of what the band represent. Sham, Control, 10uM ECH A should be typed on the bands.

Author Response

Major comments

Point 1: Since the author explain the main mechanism is driven by reducing the cell death regulation pathway it would be important to quantify the ischemic-induced apoptosis from the infarcted brain region and show the effect of EchA form the same region. I recommend measuring the fragmentation of nuclear DNA of the penumbral area by TUNEL (terminal deoxynucleotidyl transferase dUTP nick-end labeling).

Response 1: Thank you for your careful recommendation. In the previous published paper, we established that alteration of ischemic-induced cell apoptosis in  rat MCAo model using Salvia miltiorrhiza Bunge, a traditional Chinese medicine. To identify the effect of the substance on apoptosis of treated cells in infarct region, we examined the protein expression levels of the apoptosis-related factors such as Bcl-2, Caspase-3 and Bax in identical way of this paper [1]. Furthermore, we are so sorry that there is not enough time to perform other experiments in deadline.

[1] Kim, R.; Lee, S.; Lee, C.Y.; Yun, H.; Lee, H.; Lee, M.Y.; Kim, J.; Jeong, J.Y.; Baek, K.; Chang, W. Salvia miltiorrhiza enhances the survival of mesenchymal stem cells under ischemic conditions. J. Pharm. Pharmacol. 2018, 70, 1228-1241.

Minor comments

Point 1: Typing error in 3.2.2 Middle Cerebral Artery Occlusion

External carotid artery (CCA) a External carotid artery (ECA)

Response 1: Thank you for your careful comment. The suggestion has been well taken (page 4 line 141).

Point 2: In figure 2. A, it is hard to visualize a difference between affected, unaffected and simultaneous in different groups. I recommend the percentage typed in each column so each column can be better compared.

Response 2: Thank you for your advice. The suggestion has been well taken. We revised Figure 2 (A) in this paper (page 8 Figure 2).

Point 3: Figure 2, legend need indication of # and **.

Response 3: Thank you for your comment. The suggestion has been well taken. We revised the figure legend of Figure 2 (page 8 line 279-281).

Point 4: In Figure 3, each western and northern blot should have an indication of what the band represent. Sham, Control, 10uM ECH A should be typed on the bands.

Response 4: Thank you for your advice. The suggestion has been well taken. We revised Figure 3 in our paper (page 9 Figure 3).

Round 2

Reviewer 2 Report

The manuscript is acceptable for journal

Author Response

The manuscript is acceptable for journal

I appreciate that you have agreed to our paper.

Reviewer 3 Report

Minor comments

Point 1: In figure 2 B the image show the statistical significance between sham and control, sham and 10uM Ech A is #. However, the legend indicate the statistical significance is ##. This should be corrected. 

Point 2: Typing error in Legend of figure 2  

(##p< 0.01, compared with the values of the sham group; **p < 0.01, compared with the values of the control group)

p should be in capital letter

Author Response

Point 1: In figure 2 B the image show the statistical significance between sham and control, sham and 10uM Ech A is #. However, the legend indicate the statistical significance is ##. This should be corrected. 

Thank you for your careful comment. The suggestion has been well taken (page 8 line 279, ##→#).

Point 2: Typing error in Legend of figure 2  

(##p< 0.01, compared with the values of the sham group; **p < 0.01, compared with the values of the control group)

p should be in capital letter

Thank you for your careful recommendation. The suggestion has been well taken (page 8 line 279-280, p→P).